# Single-cell analysis reveals the stromal dynamics and tumor-specific characteristics in the microenvironment of ovarian cancer

Linan Zhang [1,2,3,10], Sandra Cascio [2,4,5,10], John W. Mellors[6], Ronald J. Buckanovich[2,4,7] & Hatice Ulku Osmanbeyoglu [1,2,8,9] ✉

High-grade serous ovarian carcinoma (HGSOC) is a heterogeneous disease, and a high-stromal/desmoplastic tumor microenvironment (TME) is associated with a poor outcome. Stromal cell subtypes, including fibroblasts, myofibroblasts, and cancer-associated mesenchymal stem cells, establish a complex network of paracrine signaling pathways with tumor-infiltrating immune cells that drive effector cell tumor immune exclusion and inhibit the antitumor immune response. In this work, we integrate single-cell transcriptomics of the HGSOC TME from public and in-house datasets ($n = 20$) and stratify tumors based upon high vs. low stromal cell content. Although our cohort size is small, our analyses suggest a distinct transcriptomic landscape for immune and non-immune cells in high-stromal vs. low-stromal tumors. High-stromal tumors have a lower fraction of certain T cells, natural killer (NK) cells, and macrophages, and increased expression of CXCL12 in epithelial cancer cells and cancer-associated mesenchymal stem cells (CA-MSCs). Analysis of cell-cell communication indicate that epithelial cancer cells and CA-MSCs secrete CXCL12 that interacte with the CXCR4 receptor, which is overexpressed on NK and CD8+ T cells. Dual IHC staining show that tumor infiltrating CD8 T cells localize in proximity of CXCL12+ tumor area. Moreover, CXCL12 and/or CXCR4 antibodies confirm the immunosuppressive role of CXCL12-CXCR4 in high-stromal tumors.

[1] Department of Biomedical Informatics, University of Pittsburgh School of Medicine, Pittsburgh, PA 15206, USA. [2] UPMC Hillman Cancer Center, University of Pittsburgh, Pittsburgh, PA 15232, USA. [3] Department of Applied Mathematics, School of Mathematics and Statistics, Ningbo University, Ningbo, Zhejiang 315211, China. [4] Magee-Womens Research Institute, Pittsburgh, PA 15213, USA. [5] Division of Gynecologic Oncology, Department of Obstetrics, Gynecology, and Reproductive Sciences, University of Pittsburgh School of Medicine, Pittsburgh, PA 15213, USA. [6] Division of Infectious Diseases, Department of Medicine, University of Pittsburgh School of Medicine, Pittsburgh, PA 15213, USA. [7] Division of Hematology/Oncology, Department of Medicine, University of Pittsburgh School of Medicine, Pittsburgh, PA 15232, USA. [8] Department of Bioengineering, University of Pittsburgh School of Engineering, Pittsburgh, PA 15219, USA. [9] Department of Biostatistics, University of Pittsburgh School of Public Health, Pittsburgh, PA 15261, USA. [10] These authors contributed equally: Linan Zhang, Sandra Cascio. ✉email: osmanbeyogluhu@pitt.edu

Ovarian cancer (OvCa) is a common gynecologic malignancy with high mortality, although the outcome of patients with OvCa has improved over time. Platinum and taxane chemotherapy has been the standard of care for three decades. Although immunotherapy could improve long-term survival, only 10–15% of patients treated with single-agent immunotherapy have a favorable clinical outcome[1]. Therefore, it is important to determine why immunotherapy is not effective in some patients and to develop rational therapeutic strategies to overcome resistance to treatment.

Mechanisms intrinsic or extrinsic to tumors can drive resistance to therapy. For example, the heterogeneity of the OvCa tumor immune microenvironment (TIME) is important in determining the response to therapy. OvCa patients with an immune-infiltrated molecular signature have a better prognosis than patients with a stromal signature[2,3]. Stromal cell subtypes, including cancer-associated mesenchymal stem cells (CA-MSCs), fibroblasts, and myofibroblasts, establish a complex network of paracrine signaling pathways with tumor-infiltrating immune cells that inhibit the antitumor immune response[4]. Highly stromal human cancers, in which $CD8^+$ T cells are localized primarily to the peritumoral stroma, respond poorly to anti-PD1/PD-L1 immunotherapy[5–7]. Understanding the complexity of the interactions between cancer cells and their microenvironment, including immune and stromal cells, is important for elucidating the mechanisms of drug resistance, identifying new molecular targets, and developing more effective OvCa treatments.

Transcriptomic profiles can identify molecular subtypes of high-grade serous ovarian cancer (HGSOC)[2,8]. Patients with desmoplastic/mesenchymal tumors have the worst prognosis, whereas patients with a high degree of immune infiltration have the best prognosis. However, bulk RNA sequencing (RNA-seq) gives only an average value for gene expression across all cells and does not provide the contribution of each cellular subset. In contrast, single-cell genomics provides cell-type-specific information on pathological changes in cancer and other diseases. For example, single-cell RNA-seq (scRNA-seq) is a powerful tool to interrogate tumor composition, revealing cellular heterogeneity and pathways at single-cell resolution. Likewise, spatial transcriptomics provides a transcriptional profile of cells in their native context[9–12] and information on how location affects cell types. These innovative technologies provide biological and therapeutic insights for cancers, including OvCa. In HGSOC ascites specimens, scRNA-seq showed that inhibition of the JAK/STAT pathway has potent antitumor activity[13]. The cellular composition of infiltrated, excluded, and desert tumor immune phenotypes have also been characterized[14]. A recent report described the role of certain stromal cell phenotypes in the regulation of the TME in HGSOC, including TGFβ-driven cancer-associated fibroblasts (CAF), lymphatic endothelial cells, and mesothelial cells[15]. A scRNA-seq study described the heterogeneity in the cell-of-origin for HGSOC tumors and revealed that a high epithelial-mesenchymal transition (EMT) subtype was associated with a poor prognosis[16]. In fact, the spatial interactions between cell clusters may influence chemo-responsiveness more than cluster composition alone[3]. Similarly, Ferri-Borgogno et al. found increased apolipoprotein E-LRP5 cross-talk at the stroma-tumor interface in OvCa tumor tissues from short-term survivors compared with long-term survivors[17].

Here, we combined in-house and public scRNA-seq datasets to create a comprehensive single-cell atlas of the HGSOC TME. We determined the effect of a high-stromal TME on cell-type-specific regulatory pathways, such as alterations in cytokines, surface receptors, signaling proteins, and transcription factors (TF) and their interconnections. We also analyzed an ovarian cancer spatial transcriptomics dataset, and we performed in vitro and ex-vivo experiments using non-immune cells and immune cells freshly isolated from HGSOC samples to validate our integrative scRNA-seq analysis. We focused on HGSOC, the most common histologic type, which is responsible for approximately 80% of all OvCa deaths[18]. The features that are associated with high- and low-stromal HGSOC can inform therapeutic strategies to improve treatment outcomes for patients with these phenotypes.

## Results

**A single-cell atlas for treatment-naïve, high- and low-stromal high-grade ovarian serous cancers**. We performed scRNA-seq to characterize the malignant, immune, and stromal cells associated with two treatment-naïve HGSOC samples—one high-stromal and one low-stromal according to stromal cell-type abundance (see MATERIALS AND METHODS and below). To increase statistical power and to ensure the generalizability of our results, we integrated our dataset with three public, treatment-naïve, primary HGSOC scRNA-seq datasets from Regner et al. [19] ($n = 2$), Xu et al. [20] ($n = 7$), and Hornburg et al. [14] ($n = 9$) yielding 106,521 cells from 20 treatment-naïve primary HGSOC tumors (Fig. 1a). Sample and dataset information is summarized in Table S1. We performed principal component analysis (PCA) using the top 2000 variably expressed genes across all 106,521 cells. For the top 30 principal components (PCs), we classified cells into transcriptionally distinct clusters using graph-based clustering (Fig. 1b, Fig. S1a). The clustering of scRNA-seq samples based on nearest neighbors did not match clustering by patient or by study, suggesting successful mitigation of batch effects.

Using the integrated scRNA-seq data and canonical marker genes, we quantified 13 coarse cellular lineages and constructed a cell-type map (Fig. 1c, d). Immune cell types included macrophages, dendritic cells (DCs), $CD4^+$ and $CD8^+$ T cells, B cells, natural killer (NK) cells, and NK $CD3^+$ cells (NK T). Non-immune cells included epithelial cancer cells (ECCs), endothelial cells, and stromal cells, which had subcategories of fibroblasts, myofibroblasts, and CA-MSCs. Interestingly, we identified a cell subpopulation defined as cancer stem cells with a transcriptomic phenotype similar to ECCs and CA-MSCs (*EPCAM*, *MUC1*, *PROM1*[21], *ALDH1A3*) (Fig. S1b, c). The stromal cell subclusters had both common and distinctive transcriptomic profiles (Fig. S1d, e). In line with previous findings, immune and stromal cells were clustered by cell identity rather than patient origin. Thus, the core HGSOC atlas integrated the 106,521 single cells, which were annotated to 13 coarse cell types, including 36,869 epithelial cells, 54,936 immune cells, 13,324 stromal cells, and 1392 endothelial cells.

We divided the samples into a high-stromal group comprising those that were densely populated with fibroblasts and CA-MSCs and a low-stromal group comprising those sparsely populated with fibroblasts and CA-MSCs according to cell-type abundance (Fig. 1e, f). Compared to high-stromal tumors, low-stromal tumors had a higher proportion of immune cells (Fig. S2a), specifically $CD8^+$ T cells ($p$-value < 0.01, Wilcoxon signed-rank test) and NK cells ($p$-value < 0.05) (Fig. 1g). However, apparent biological differences in cell-type proportions between individual samples and groups can be caused by differences in cell isolation protocols and scRNA-seq platforms. After performing dimension reduction using PCA on cells by cell type, we observed larger variations for PC1 and PC2 among high- and low-stromal tumors for major cell types, including CA-MSCs, fibroblasts, and ECCs (Fig. S2b), indicating cell-type-specific gene expression differences between high- and low-stromal tumors.

To determine whether the proportion of stromal cells affects disease progression and outcome, we analyzed bulk

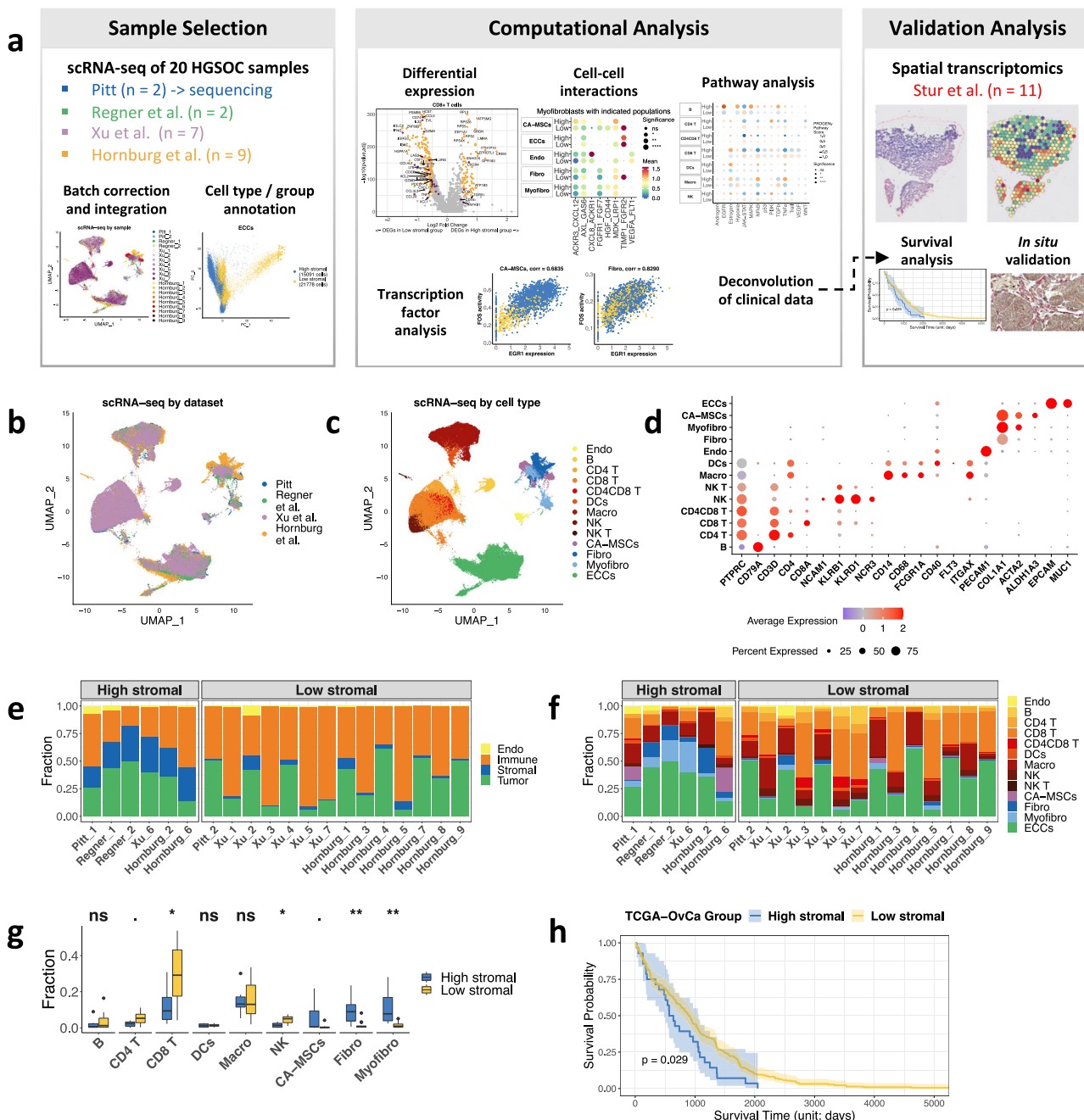

**Fig. 1 Characterization of endothelial, immune, stromal, and tumor cell types in HGSOC. a** Summary of the data integration and analysis workflow. **b** Uniform manifold approximation and projection (UMAP) clustering of the integrated scRNA-seq data, colored by dataset and **c** by cell type. Each dot represents a single cell. **d** Dot plot showing the expression levels of selected canonical cell markers that were used to identify the clusters for each cell type. **e** Stacked bar plots showing the fraction of endothelial, immune, stromal, and tumor cell types and **f** the refined cell types for all samples. **g** Box plot showing the fractions of selected cell types in each sample, colored by tumor group (high-/low-stromal groups). The *p*-values are computed from the two-sided Wilcoxon signed-rank test between the two groups for each cell type. Statistical significance is coded by the following symbols. *p*-value < 0.1, * *p*-value < 0.05, ** *p*-value < 0.01, and *** *p*-value < 0.001. **h** Kaplan–Meier survival curve for TCGA-OvCa samples split by the high-/low-stromal groups. The high- and low-stromal tumors patients were defined by the expression of *COL1A1*.

transcriptomic data from 248 OvCa samples profiled by the Cancer Genome Atlas Program (TCGA)[8]. With the scRNA-seq data as a reference, we used the bMIND deconvolution algorithm[22] to infer the proportion of epithelial, endothelial, immune, and stromal cells for each sample from the bulk RNA-seq data (Fig. S2c). Based on the correlation between the proportion of each cell type with patient-matched overall survival, we found that a high abundance of stromal cells was associated

with a lower survival rate for OvCa patients (*p*-value = 0.029, log-rank test) (Fig. 1h).

**Stromal features associated with high- and low-stromal TME.** Analysis of the patterns of differentially expressed genes (DEGs) for ECCs, CA-MSCs, fibroblasts, and myofibroblasts revealed that several cytokines and surface proteins (SPs) were differentially

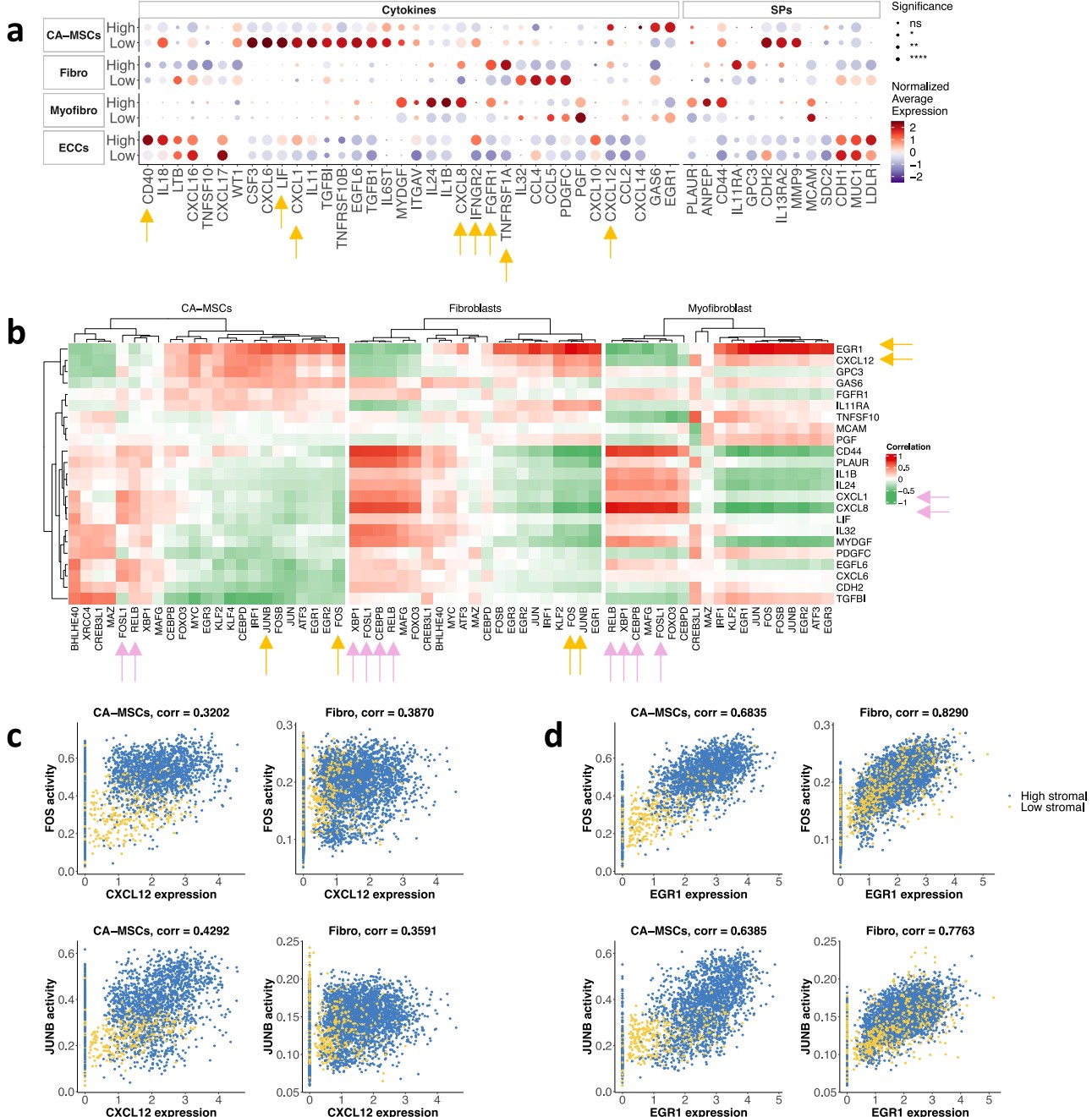

**Fig. 2 Stromal features associated with high- and low-stromal TIME. a** Average expression of selected cytokines and surface proteins (SPs) in each cell type and tumor group. Dot plot showing the mean gene expression *z*-score of differentially expressed cytokines and SPs across cell types. The expression of each gene is normalized to a mean value of zero and unit standard deviation. Expression values are indicated by color. Adjusted *p*-values for each differentially expressed gene between high- and low-groups are indicated by circle size. Statistical significance is coded by the following symbols. *p*-value < 0.1, \**p*-value < 0.05, \*\**p*-value < 0.01, and \*\*\**p*-value < 0.001. **b** Heatmap revealing correlations between inferred TF activities (columns) and SP/cytokine expression (rows) in CA-MSCs, fibroblasts, and myofibroblasts. For clarity, we selected highly correlated SP/cytokine-TF pairs (Table S4) and used the union of the selected genes. The correlation coefficients of the top correlated TFs with each SP/cytokine are shown for each cell type. **c, d** Scatter plots of selected highly correlated gene-TF pairs for each cell type. Each dot represents a cell colored by the tumor group.

expressed between high- and low-stromal tumors (Fig. 2a, Fig. S3). The expression of *CD40* and *TNFRSF1A*, both members of the TNF-receptor superfamily; *CXCL1*, a factor known to induce myeloid cell recruitment[23,24]; *IFNGR2*; and *LIF* were higher in ECCs of high-stromal tumors than low-stromal tumors. Compared to low-stromal tumor samples, CA-MSCs, fibroblasts, and myofibroblasts from high-stromal samples showed higher expression of immunosuppressive cytokines, including *CXCL8*

and *CXCL12*, and the fibroblast growth factor receptor *FGFR1*, which is involved in tumor growth and angiogenesis[25] (Fig. 2a, Fig. S3a).

We analyzed cell-type specific TF activity using SCENIC[26,27] and characterized the associations between inferred TF activity and cytokine/SP expression. For each cell type, we computed Spearman correlation coefficients (Rho) for each TF and cytokine/SP pair, yielding two-way clustering for these pairwise

correlations in CA-MSCs, fibroblasts, and myofibroblasts (Fig. 2b). We identified both novel and known relationships for each cell type. Activities of CEBPB, FOSL1, RELB, and XBP1 correlated highly with expression of *CXCL1* and *CXCL8* in fibroblasts and myofibroblasts (Fig. S4a, b); activities of FOSL1 and RELB also correlated with expressions of *CXCL1* and *CXCL8* in CA-MSCs (Fig. S4c). This is consistent with previous studies reporting that NF-κB (NFKB1) and RELB interact with CAAT enhancer binding protein (C/EBP) elements modulating *CXCL8* transcription[28]. Importantly, the analysis suggests that cytokines and SPs couple with shared, context-dependent downstream TFs in various cell types associated with stromal cell infiltration. For example, FOS and JUNB correlated highly with the expression of *CXCL12* and *EGR1* in CA-MSCs and fibroblasts (Fig. 2c, d). In a similar analysis of ECCs (Fig. S5a), we found that ECCs from high-stromal tumors had higher levels of ATF5, RELB, and TRIM28 that correlated with *CXCL1* expression (Fig. S5b). In contrast, levels of BCL3, HIVEP1, and ZBTB7A were higher in ECCs of low-stromal tumors and correlated with the expression of *MUC1* (Fig. S5c). We found elevated levels of EGFR, estrogen, MAPK, PI3K, TGFβ, VEGF, and WNT signaling in ECCs in high-stromal tumors compared to low-stromal tumors (adjusted $p$-value $< 10^{-20}$, Wilcoxon signed-rank test) (Fig. S5d, e) and elevated levels of JAK-STAT and p53 in ECCs and fibroblasts in low-stromal tumors compared to high-stromal tumors (adjusted $p$-value $< 10^{-20}$) (Fig. S5d).

**Stromal cells modulate the phenotypes of innate immune cells.** Using a murine model, we showed previously that CA-MSCs modulate gene expression in macrophages, inducing immunosuppression[4]. Here, we evaluated the effect of stromal cells on various human myeloid cell types. Macrophages and DCs from high- and low-stromal tumor tissues showed different gene expression signatures (Fig. 3a, Fig. S3b). *IL1RN* and *CCL20* expression was higher in macrophages from high-stromal vs. low-stromal tumors, whereas *NFKB1* and *CXCL8* were upregulated in both macrophages and DCs from high-stromal tumors compared to low-stromal tumors. Conversely, both macrophages and DCs infiltrating high-stromal tumors showed lower expression of *CXCL9* and *CXCL10*, two cytokines that contribute to the generation of a "hot" tumor microenvironment[29] (Fig. 3b, Fig. S6a). Further, macrophages infiltrating high-stromal tumors displayed high levels of chemokines *CXCL1*, *CXCL5*, and *IL1A*, which suppress anti-cancer immunity[30] (Fig. 3c, Fig. S6b). DCs infiltrating high-stromal tumors showed upregulation of *CXCL2*, *CXCL3*, *CXCL8*, *NFKB1*, and *VEGFA* compared to DCs infiltrating low-stromal tumors (Fig. 3d, Fig. S6c).

To confirm that CA-MSCs reprogram myeloid cells, we evaluated the protein expression of 80 cytokines and chemokines from the supernatant of macrophages cultured with the conditioned medium (CM) of OVCAR3 tumor cells (TC), ovarian MSC cultured alone (MSC), or OVCAR3 cells co-cultured with MSC (TC/MSC) as described in Materials and Methods. We found that the addition of MSCs to a tumor cell culture induced the expression of chemokines *CXCL1*, *CXCL5*, and *CXCL13* (Fig. 3e, f). These cytokines are known to induce proliferation, migration, and metastasis of cancer cells[31,32].

We also determined pairwise TF-cytokine/SP correlations for innate immune cells (Fig. S7a). Levels of CEBPB, ETS2, FOSL1, and NFE2L2 correlated highly with expression of *CXCL2*, *CXCL8*, *CXCL16*, *IL1B*, *PLAUR*, and *VEGFA* in DCs (Fig. S7b). Levels of BCL3, CEBPB, ETS2, NFE2L2, NFKB1, and XBP1 correlated highly with expressions of *CXCL2*, *CXCL3*, *CXCL8*, *IL1B*, and *PLAUR* in macrophages (Fig. S7c). This is consistent with previous reports that CEBPB regulates *CXCL1* expression[28],

whereas NF-κB is an essential modulator of transcription of *CXCL-1/-2/-3/-8*[33].

**Stromal cells modulate the phenotype of adaptive immune cells.** Adaptive immune cells infiltrating high- and low-stromal tumors showed different gene signatures (Fig. 4a, Fig. S3c). Interestingly, *CXCR4*, the receptor of CXCL12 and one of the immunosuppressive cytokines upregulated in the stromal cells of high stroma tumors, was upregulated in CD8$^+$ T and NK cells of high-stromal samples. EGR4, FOSL2, GTF2B, and NFKB1 levels correlated with the expression of *CXCR4* in CD8$^+$ T and/or NK cells (Fig. 4b–d). ETV7 and IRF2 levels correlated with the expression of *GZMA*, *GZMB*, and *IL2RG* genes that are associated with CD8$^+$ T and NK cell activation (Fig. S8a). Levels of RUNX3 and IRF2 correlated with expressions of *CCL4*, *CCL5*, *GZMA*, and *GZMB* in CD8$^+$ T cells (Fig. S8b). In NK cells, levels of ETV7, IRF2, IRF7, IRF9, STAT1, and STAT2 correlated with expressions of *FASLG*, *GZMA*, *LAG3*, and *TNFSF10* (Fig. S8c).

Using gene expression data from adaptive immune cells, we analyzed cell-specific pathway enrichment to determine whether a high- vs. low-stromal TME affected similar or different molecular pathways (Fig. S5e). High-stromal tumors showed high levels of (1) estrogen signaling in macrophages, DCs, NK, CD4$^+$ T, and CD8$^+$ T cells; (2) MAPK in B cells, NK, macrophages, and T cells; and (3) TNF and NK-KB in DCs and macrophages (all with adjusted $p$-value $< 10^{-20}$, Wilcoxon signed-rank test). In particular, estrogen and JAK-STAT pathways were up- and down-regulated, respectively, in most immune cells of high-stromal tumors. As expected, TNFα (a marker of the activated adaptive response) was down-regulated in CD4$^+$ T, CD8$^+$ T, and NK cells (Fig. S5e).

**The CXCL12–CXCR4 axis reduces the cytotoxic activity of CD8$^+$ T cells and NK cells in high-stromal tumor samples.** Communication among the many TME components plays a critical role in tumor progression and treatment response. Using the CellPhoneDB ligand–receptor complexes database[34], we determined the interactions between immune cells and non-immune cells in high- and low-stromal tumors. We found that both CA-MSC-secreted and fibroblast-secreted *CXCL12* interacted with the *CXCR4* receptor on CD8$^+$ T and NK cells (Fig. 5a, b), and we saw over-expression of *CXCL12* in CA-MSCs and fibroblasts in high-stromal tumors (Fig. 5c, Fig. S9a). Consistent with those results, we found that *CXCR4* was overexpressed in CD8$^+$ T and NK cells of high-stromal tumors (Fig. 5d, Fig. S10b).

CXCL12 induces immune suppression in the TME by sequestering CD8$^+$ T cells in the tumor stroma, away from tumor islets, and it induces the accumulation of myeloid-derived suppressor cells[35–37]. When we measured the expression of markers associated with activated CD8$^+$ T and NK cells, we found that *CXCR3*, Granzyme B (*GZMB*), Interferon Gamma (*IFNG*), and *IL2R* were significantly upregulated in low-stromal samples in both cell populations (Fig. 5e, f; Fig. S10a, b). Moreover, *NCR3* (NKp30), a receptor associated with NK cell activation, was upregulated in low-stroma tissues (Fig. 5f). To confirm the immune suppressive activity of CA-MSC-secreted *CXCL12*, we performed an in vitro assay. We cultured CD8$^+$ T cells, freshly isolated from peripheral blood mononuclear cells (PBMC), with the CM of CA-MSCs that were isolated from human HGSOC ascites. To determine the inhibitory activity of *CXCL12*, CD8$^+$ T cells were treated with recombinant protein *CXCL12* (Fig. 5g). CA-MSC CM significantly reduced the secretion of *GZMB* and *IFNG*. However, the addition of *CXCR4* antibody (Ab) to CA-MSC CM restored *GZMB* and *IFNG* secretion.

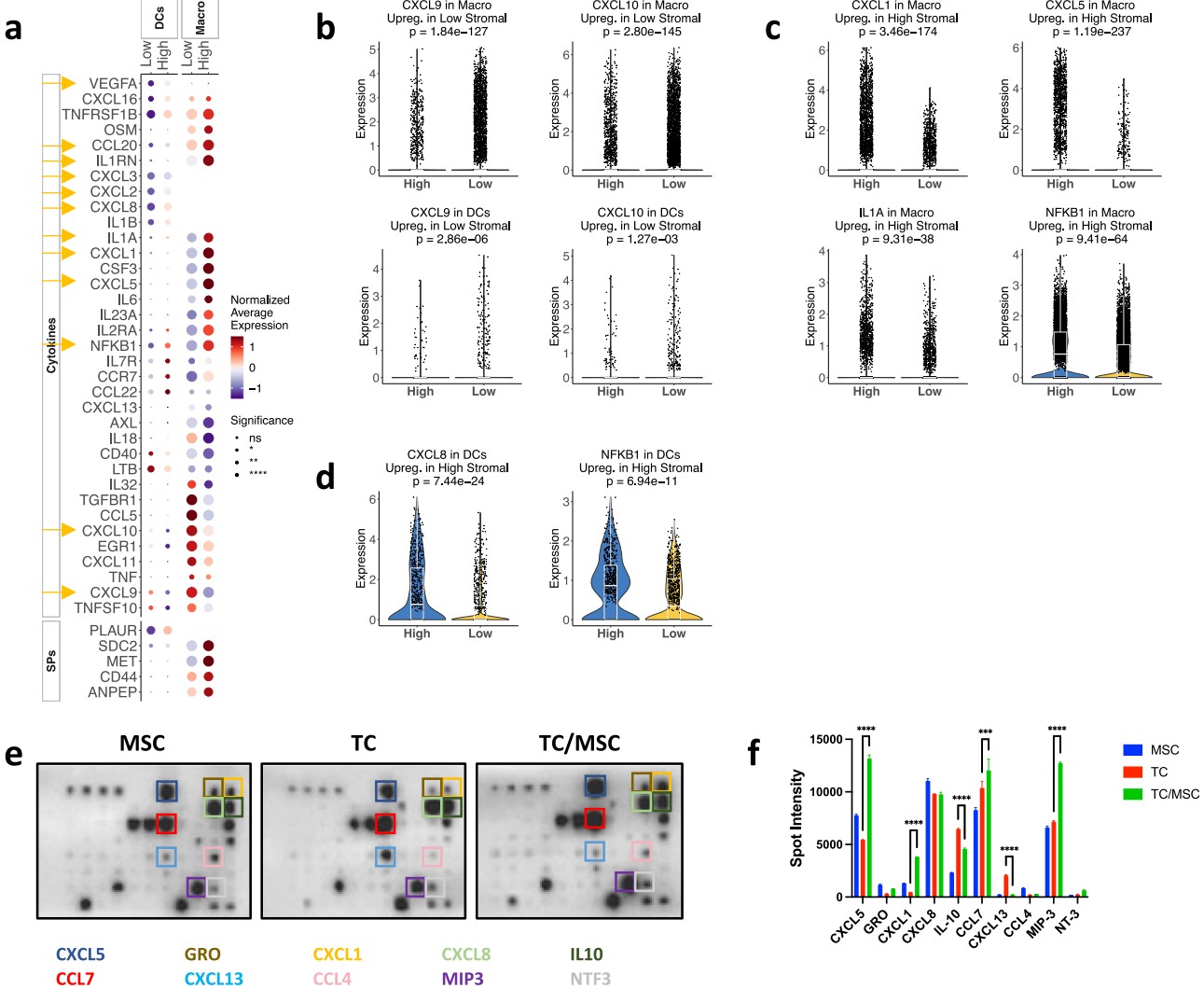

**Fig. 3 Impact of stromal cells on myeloid cell-secreted cytokines. a** Average expression of selected cytokines and SPs in each cell type and tumor group. Dot plot showing the mean gene expression z-score of differentially expressed cytokines and SPs across different cell types. The expression of each gene is normalized to a mean value of zero and unit standard deviation. Expression values are indicated by color. Adjusted p-values for each differentially expressed gene between high- and low-groups are indicated by circle size. **b** Violin plots showing the expression of *CXCL9* and *CXCL10* in macrophages and dendritic cells (DCs). **c** The expression of *CXCL1*, *CXCL5*, *IL1A*, and *NFKB1* in macrophages, and **d** the expression of *CXCL8* and *NFKB1* in DCs by high-/low-stromal group. *P*-values are computed from the two-sided Wilcoxon test, adjusted based on Bonferroni correction using all genes in the dataset. Statistical significance is coded by the following symbols. *p*-value < 0.1, *p*-value < 0.05, **p*-value < 0.01, and ***p*-value < 0.001. **e** Top, human cytokine arrays used macrophage supernatants cultured with mesenchymal stem cells (MSC), tumor cells (TC), or MSC/TC conditioned media as described in Material and Methods. **f** Bar plot summarizing relative intensities of cytokines with the greatest increase, plotted in Image J.

To validate the findings based on the OvCa spatial transcriptomics dataset from Stur et al. [3], we divided eleven samples into high-stromal (blue, with high expression of *COL1A1* and *ACTA2*) and low-stromal (yellow, with low expression *COL1A1* and *ACTA2*) samples (Fig. 6a). We identified features whose variability in expression could be explained to some degree by spatial location via Moran's *I* test, which showed a strong spatial autocorrelation for *CXCL12* across all samples (Fig. 6b). *CXCL12* was highly expressed in the stromal area (indicated by higher expression of *COL1A1* and *ACTA2*) (Fig. 6c, d; Fig. S11). Interestingly, both CD8+ T and NK cells were also in close proximity to the *CXCL12*+ *ACTA2*+ tumor area (Fig. 6c, Fig. S11a) in high-stromal samples, which suggested that *CXCL12* could keep CD8+ T and NK cells from trafficking into the tumor islets. In low-stromal tumors, *CXCL12* was expressed at low levels, and immune cells infiltrated the tumor area (Fig. 6d, Fig. S11b). Additionally, we computed the Spearman correlation

between expression of *CXCL12* and *COL1A1* and *ACTA2* across samples, and we observed a positive correlation between *CXCL12* and *COL1A1* and *ACTA2* (Fig. 6f). Our immunohistochemistry assay further confirmed that in OvCa tissues, *CXCL12* was expressed mainly in stromal cells and co-localized with CD8+ T cells (Fig. 6f).

## Discussion

Here, we identified the molecular signatures of the TME of high- vs. low-stromal OvCa tumors. We hypothesized that a systems biology approach to characterizing the TIME in these two tumor types would identify differences that could be used for the development of combinatorial targeted therapy and immunotherapies. We combined in-house and public scRNA-seq, spatial transcriptomics data, and our immunological assays to define the OvCa TIME and characterize the biology underlying the low- and high-stromal phenotypes. We found that high- and

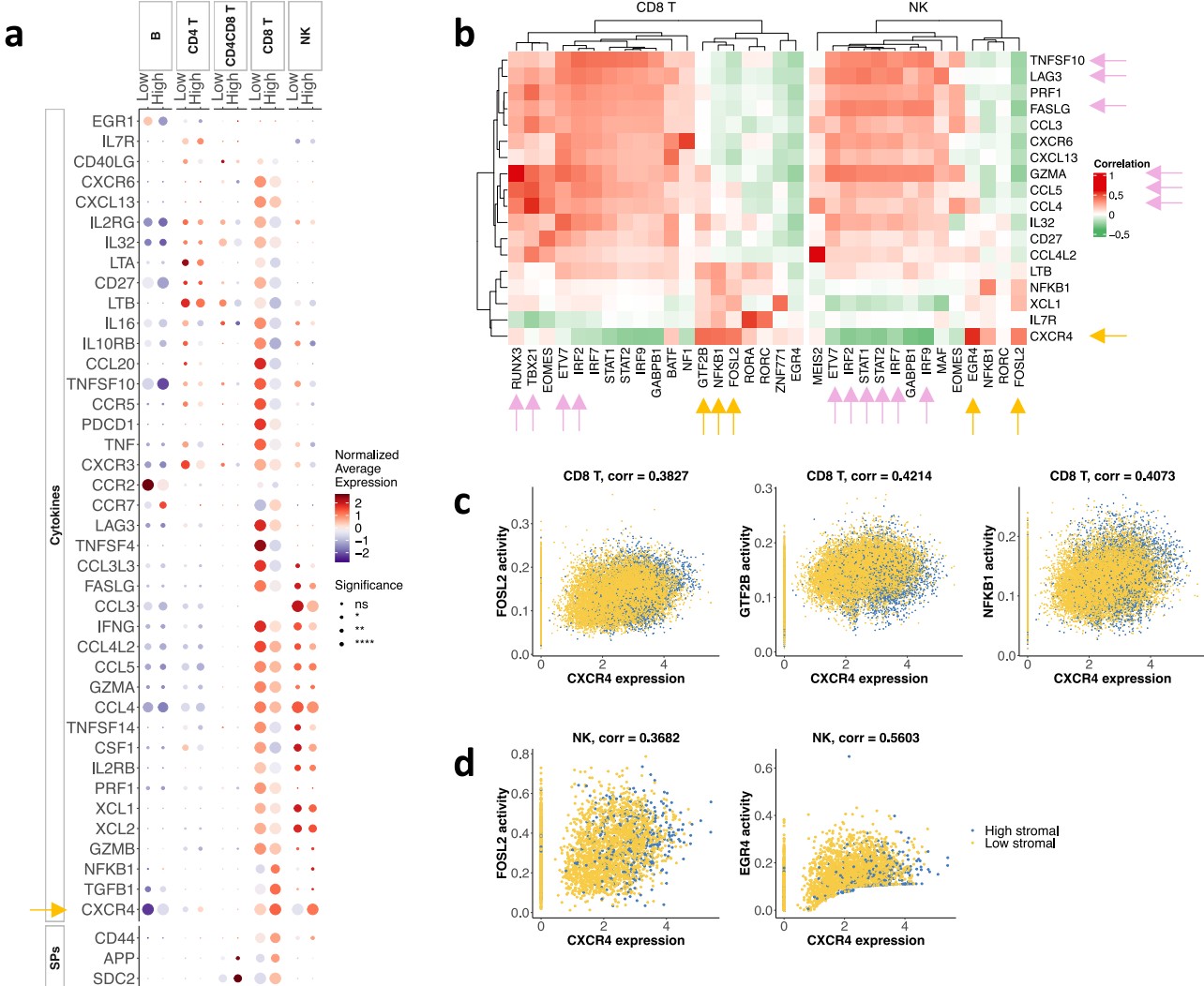

**Fig. 4 Stromal cells modulate the phenotype of adaptive immune cells. a** Average expression of selected cytokines and SPs in each cell type and tumor group. Dot plot showing the mean gene expression z-score of differentially expressed cytokines and SPs across different cell types. The expression of each gene is normalized to a mean value of zero and unit standard deviation. Expression values are indicated by color. Adjusted p-values for each differentially expressed gene between high- and low-groups are indicated by circle size. Statistical significance is coded by the following symbols. p-value < 0.1, *p-value < 0.05, **p-value < 0.01, and ***p-value < 0.001. **b** Heatmap revealing correlations between inferred TF activities (columns) and SP/cytokine expression (rows) in CD8+ T cells and NK cells. For clarity, we selected highly correlated SP/cytokine-TF pairs (Table S4) and used the union of the selected genes. Correlation coefficients of the top correlated TFs with each SP/cytokine are shown for each cell type. **c** Scatter plots of selected highly correlated TF-*CXCR4* pairs for CD8+ T cells and **d** for NK cells. Each dot represents a cell, colored by tumor group.

low-stromal tumors present very different immune landscapes. Stromal cells regulate innate and adaptive immune responses, thereby enhancing their immunosuppressive properties. In particular, the immunosuppressive CXCL12–CXCR4 cell signaling axis is prominent in high-stromal tumors.

We showed previously that stromal cells, particularly CA-MSCs, inhibit the response to anti-PD-L1 therapy in an immunotherapy-responsive syngeneic mouse model of OvCa[4]. Moreover, in our mouse model, scRNA-seq analysis revealed that myeloid cells mediate the resistance to immunotherapy-induced stromal cells[4]. Consistent with those results, here we found that in human OvCa tissues, infiltrating myeloid cells, including macrophages and DCs, were modulated by stromal cells. In high-stromal tumors, our scRNA-seq and cytokine array analyses revealed that myeloid cells expressed high levels of *CXCL1*, *CXCL5*, and *CXCL13* chemokines. *CXCL1* and *CXCL5* are known to (a) recruit myeloid-derived suppressor cells (MDSC) into the TIME, which suppress T cells and NK cells[30,38], and (b) promote

migration and metastasis of TC[39]. Previous studies have shown that MSCs induce the recruitment of macrophages into the tumors and convert them into an M2-like signature via secretion of inflammatory factors, including IDO, IL-10, IL-6 PEG2, TGF-beta, and TNF-alpha[40,41]. The expression of *CXCL13* is mainly induced by TNF-a and IL-10[42,43], whereas *CXCL1* and *CXCL5* expression levels are modulated by TNF-a[32,39]. Therefore, a potential mechanism of CA-MSCs to induce the expression of *CXCL1*, *CXCL5*, and *CXCL13* in macrophages is via the secretion of cytokines and chemokines, including TNF-alpha, IL-10, and TGF-beta. We also found that in low-stromal tumors, myeloid cells secreted high levels of *CXCL9* and *CXCL10*, factors associated with tumor-infiltrating CD8+ T cells[44] and a positive response to immune checkpoint therapy[45].

Our integrative scRNA-seq analysis and dual IHC staining revealed that the CXCL12–CXCR4 axis is a key inhibitor of antitumor immunity in high-stromal OvCa samples. CXCL12 is expressed in a variety of cells in bone marrow, liver, lungs, lymph

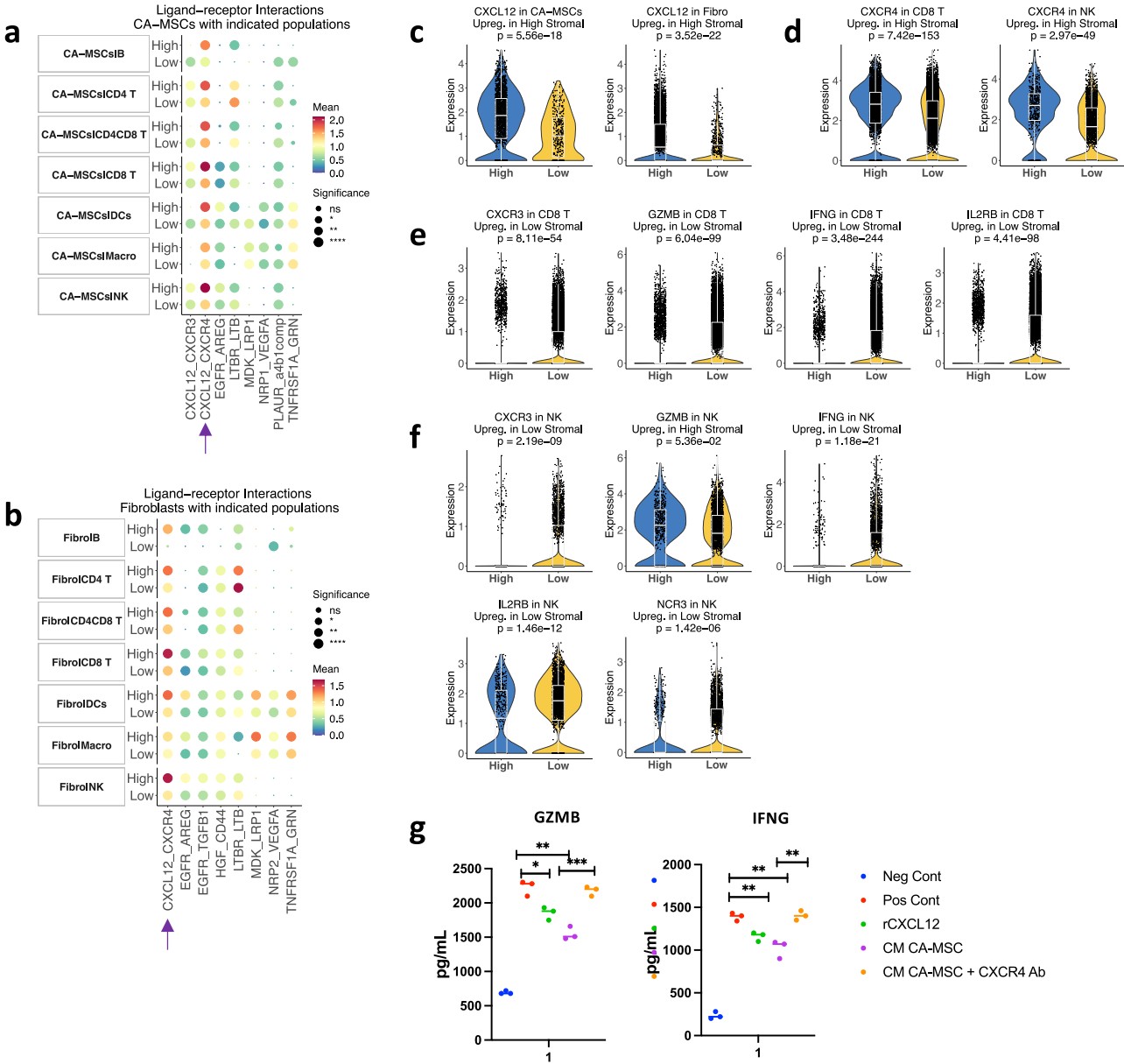

**Fig. 5 The CXCL12–CXCR4 axis reduces the cytotoxic activity of CD8$^+$ T cells and natural killer cells in high-stromal tumor samples.** Statistically significant interactions between **a** CA-MSCs, **b** fibroblasts, and other cell types using the CellPhoneDB pipeline. Size indicates *p*-values, and color indicates the means of the receptor-ligand pairs between the two tumor groups. **c** Violin plots showing the expression of *CXCL12* in CA-MSCs and fibroblasts by high-/low-stromal group. The *p*-values are computed from the two-sided Wilcoxon tests, adjusted based on Bonferroni correction using all genes in the dataset. Statistical significance is coded by the following symbols. *p*-value < 0.1, \**p*-value < 0.05, \*\**p*-value < 0.01, and \*\*\**p*-value < 0.001. **d** Violin plots showing the expression of *CXCR4* in CD8$^+$ T cells and natural killer (NK) cells by high-/low-stromal group. **e** Violin plots showing the expression of *CXCR3*, *GZMB*, *IFNG*, and *IL2RB* in CD8$^+$ T cells and **f** in NK cells by high-/low-stromal group. **g** Summary of enzyme-linked immunosorbent assay (ELISA) of *GZMB* and *IFNG* secretion in splenic a-CD3/CD28+Il-2–activated CD8$^+$ T cell (Pos Cont), recombinant CXCL12-treated cells, and CD8$^+$ T cells cultured with conditioned medium (CM) from CA-MSCs with or without anti-CXCR4 Ab. Unstimulated CD8$^+$ T cells were used as negative control (Neg Cont).

nodes, stromal cells (e.g., fibroblasts), and endothelial cells[46–48]. CXCL12 bound to its specific G protein-coupled receptor CXCR4 induces a plethora of downstream signaling events involving ERK1/2, RAS, PLC/MAPK, p38 MAPK, and SAPK/JNK, which are in turn responsible for various biological and pathological processes including angiogenesis and tumor metastasis[49]. We found that both CD8+ T cells and NK cells are abundant in the CXCL12+ tumor area. This is consistent with previous studies in pancreatic cancer showing that CAF-secreted CXCL12 attracts peripheral CXCR4+ CD8+ T cells toward activated CAFs located in the stromal regions surrounding the tumor[36,50]. This leads to

the sequestration of CD8+ T cells in the stromal compartment and reduced migration into tumor islets[36,50]. In addition, pre-clinical studies showed that both pharmacological inhibition of CXCR4 and genetic ablation of *CXCL12*-producing CAFs led to a rapid accumulation of CD8+ T cells within the tumor and reduced tumor growth[36]. Moreover, the blockade of CXCR4 alleviates tumor desmoplasia and increases T-cell infiltration in metastatic breast cancer[49]. Despite reports of the expression of CXCL12 in various tumors, there is little information on the role of the CXCL12–CXCR4 axis in the OvCa TIME and the

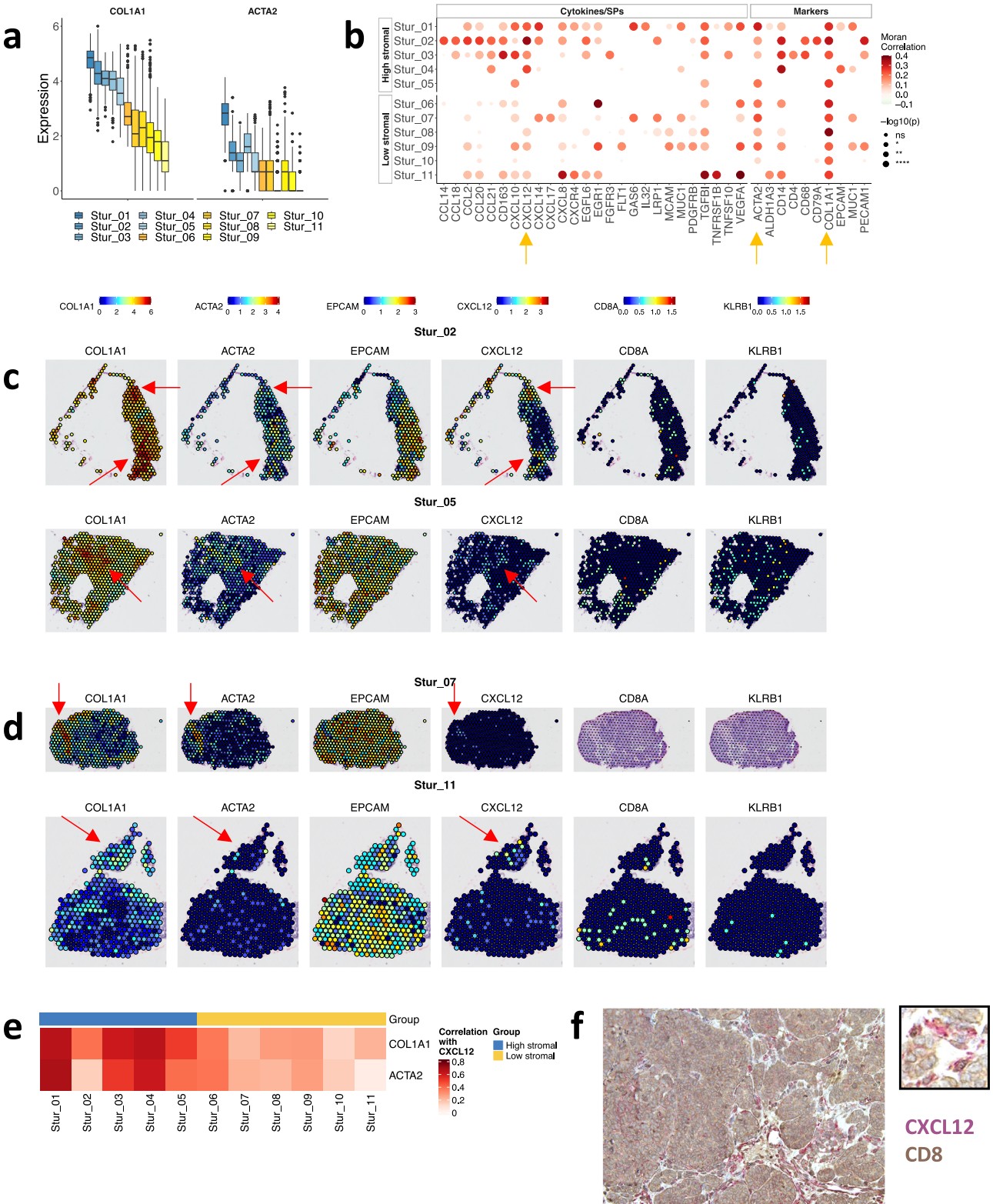

**Fig. 6 Spatial distribution of CXCL12, CD8+ T cells, and NK cells in high- and low-stromal OvCa tissues. a** Box plot showing the expression of selected marker genes by sample. Based on the average expression of *COL1A1*, the samples were split into high-stromal (blue) or low-stromal (yellow) groups. **b** The spatial autocorrelation of selected cytokines, SPs, and marker genes by sample. Moran's I test, implemented in the Seurat *FindSpatiallyVariableFeatures* function, was applied to compute the autocorrelation between the expression of each gene and its spatial location. **c** Representative spatial feature plots showing mRNA levels of stromal markers (COL1A1, ACTA2, EPCAM), CXCL12, CD8+ T cell marker (CD8A) and NK marker (KLRB1) from two high-stromal samples **d** and two low-stromal samples. Arrows indicate regions COL1A1, ACTA2 and CXCL12. **e** Heatmap showing the Spearman correlation coefficients between the expression of COL1A1/ACTA2 and CXCL12. **f** Representative image of *CD8* (brown) and *CXCL12* (magenta) double stained in a paraffin-embedded human OvCa tissue section. The scale bar equals 100 mm.

mechanisms by which this chemokine affects tumor growth and spread as well as TIME dynamics.

Our findings are consistent with OvCa molecular subtyping studies, which indicate that patients with a reactive stromal tumor subtype tend to show early relapse and short overall survival[2]. The association between many stromal cells and a poor clinical outcome has also been observed in other cancers, including prostate and pancreatic cancers[51,52], and stromal cells impact chemotherapy in OvCa[53–55]. Similarly, high desmoplasia is linked to immunosuppression[4,56]. We and others found that in syngeneic and immunogenic mouse models, CA-MSCs inhibit anti-tumor immunity via multiple mechanisms. By secreting cytokines and chemokines such as CCL2, CX3CL1, and TGFß1, CA-MSCs recruit monocytes and macrophages to the desmoplastic TME and promote their differentiation into a pro-angiogenic and immunosuppressive phenotype[4]. Moreover, CA-MSCs directly inhibit CD8$^+$ T cells and NK cell activity[4,57–59]. Importantly, stromal cells drive tumor immune exclusion by blocking CD8$^+$ T cell trafficking into the tumor islets and inducing resistance to immune therapy[4]. Similarly, OvCa patients with many T cells infiltrating the tumor islets ("hot" tumors) have a better prognosis than patients with either tumor immune exclusion, in which T cells are restricted to the tumor stroma ("immune excluded"), or with an absence of T cells in the tumor islets (immune "desert")[60,61].

Nonetheless, this study has limitations. First, the relatively small sample size imposes constraints on the flexibility of multivariable modeling. Given that our in-house cohort comprises only two tumor types, we attempted to address this limitation by integrating additional scRNA-seq datasets to capture the variability present across HGSOC tumors. To enable a joint analysis of both in-house and public datasets, we had to choose a data integration approach (for detailed information, please refer to the "Materials and methods" section). It is important to note that each cohort entails a batch effect due to slight variations in sample processing and differences in scRNA-seq library preparation and sequencing. However, the integration of several datasets has the advantage of mitigating biases related to cell preparation and dissociation, such as the preferential liberation of specific cell types during tissue dissociation. Additionally, our ligand–receptor analysis relies on a limited database of known ligand–receptor interactions and can only infer potential communication between different cell types. Therefore, the ligand–receptor analysis should be considered primarily for hypothesis generation, and a more sensible future approach would involve conducting functional studies to delve into the molecular crosstalk. Further, our study lacks matching clinical data regarding treatment allocation and responses to treatment. This limitation is due to the retrospective nature of our analysis.

In summary, we used integrative scRNA-seq and cell-cell communication mapping to show that tumors with many stromal cells, such as CA-MSCs, fibroblasts, and myofibroblasts, establish a paracrine signaling network with infiltrating immune cells. This signaling pathway results in an immunosuppressive TME. Therefore, molecular features unique to high-stromal OvCa may provide a framework for the development of novel strategies to improve treatment outcomes for this deadly gynecological cancer.

## Methods
### Single-cell RNA sequencing
*Generation of cells and scRNA-seq libraries.* We selected one high-stromal and one low-stromal treatment-naïve high-grade ovarian serous cancer (HGSOC) tumor for in-house single-cell RNA sequencing (scRNA-seq). We used fluorescence-activated cell sorting to sort immune cells (CD45$^+$) and non-immune cells (CD45$^-$) from each tumor sample. A separate single-cell library was generated for each sample from sorted live CD45$^+$ cells using the Chromium Single Cell 30 Reagent (5′ V1 and Next GEM 5′ V1.1 chemistry). Briefly, sorted cells were resuspended in phosphate-buffered saline (PBS) with 0.04% bovine serum albumin (BSA; Sigma), counted using the Cellometer Auto2000 (Nexcelom), loaded into the single-cell chip, and processed through the 10× controller to generate droplets each containing a single cell and bead. Our target was 10,000 cells per sample. Cells were lysed in the droplet, and the RNA was reverse transcribed, producing cDNA representing single-cell transcriptomes with bead-specific sequences to identify the cell type. The cDNA was isolated, amplified by 13 cycles of PCR, size-selected using SPRIselect beads, and then subjected to enzymatic fragmentation, end repair, and poly(A) tailing. Adapters were ligated, and sample indices were added by PCR. Samples were then size-selected by SPRIselect, and the concentration was determined by KAPA DNA Quantification. Our library constructs contained P5 and P7 Illumina sequencing adapters, a 16-base-pair cell barcode, a 10-base-pair unique molecular identifier (UMI), a gene insert, and an i7 sample index.

*Sequencing and raw data processing.* Two libraries per patient were pooled and sequenced on an Illumina NovaSeq 6000 platform. The CellRanger (v3.1.0) *count* command was used to align sequencing reads in fastq files to the 10× GRCh38 transcriptome (v3.0.0). Outputs of the four reads were then aggregated using CellRanger *aggr* into a single feature-barcode matrix.

*Public datasets.* Three public scRNA-seq ovarian carcinoma datasets comprising 18 OV human primary tumor samples were collected. We integrated GSE173682 by Regner et al. [19], GSE184880 by Xu et al. [20], and EGAS00001004935 by Hornburg et al. [14] datasets with our in-house scRNA-seq data ($n = 2$) (GSE232314) (see below) to create a large reference dataset. The spatial transcriptomics dataset GSE189843 by Stur et al. [3] was used for validation. The TCGA- OvCa raw count matrix containing 18,374 genes across 299 samples was downloaded from the Broad Institute of MIT & Harvard (https://gdac.broadinstitute.org/). The clinical TCGA-OvCa data containing 576 samples were downloaded using the *survivalTCGA* function from the R package RTCGA (version 1.20.0). Sample and data information are summarized in Table S1.

### Analysis of scRNA-seq data
*Data preprocessing.* The in-house filtered feature-barcode matrix (filtered_feature_bc_matrix.h5 from CellRanger *aggr*) was loaded into a Seurat object using the Seurat R package (version 4.0.1)[62]. The matrix.mtx, features.tsv, and barcodes.tsv files for each sample from the Gene Expression Omnibus (GEO) datasets[19,20] were loaded using the *Read10X* function from Seurat. The count matrix for each sample of the European Genome-phenome Archive (EGA) dataset[14] was loaded using the *read.csv* function in R. A total of 18 samples were selected from the public datasets.

*Quality control.* The gene-by-cell matrix described above was loaded as a Seurat object via the *CreateSeuratObject* function to include features detected in at least three cells and in cells with at least 400 features. The analyzed cells had a minimum of 500 expressed genes, 1000 UMI counts, and less than 30% mitochondrial gene expression. SCTransform was used for normalization[63], and DoubletFinder (version 2.0.3)[64] was used to calculate and filter cells with the doublet formation rate set to 5%. Sample matrices were merged by the patient and then

renormalized and scaled using log-normalization with a scale factor of 10,000.

*Batch correction and integration.* We scaled and centered 2000 variable features, detected with the variance-stabilizing transformation (VST) method, and regressed out the mitochondrial read percentage. Principal component analysis (PCA) was performed on the filtered feature-by-barcode matrix. The uniform manifold approximation and projection (UMAP) embeddings[65] were based on the first 30 principal components. Subsequently, data integration was performed using the R package Harmony (version 0.1.0)[66] using the function *RunHarmony* to remove the batch effects among samples. The integrated data contained 33,718 genes across 111,062 cells.

*Major cell-type identification.* Graph-based clustering was performed for integrated data using the Louvain algorithm implemented in Seurat (resolution = 2). Cells were assigned to one of the following 13 cell types based on the expression of the marker genes in each cluster: (i) Immune cells (*PTPRC*) including B cells (*CD79A*, *CD19*, *MS4A1*), CD4$^+$ T cells (*CD3D* and *CD4*), CD8$^+$ T cells (*CD3D* and *CD8A*), CD4$^+$ CD8$^+$ double-positive T cells (*CD3D*, *CD4*, and *CD8A*), dendritic cells (*CD40*, *CD1C*, and *ITGAX*), macrophages (*CD14*, *CD68*, and *FCGR1A*), and NK T cells (*CD3D$^{-/+}$*, *KLRB1, NCAM1*). (ii) Tumor cells (*EPCAM* and *MUC1*) or ECCs. (iii) Stromal cells (COL1A1), including cancer-associated mesenchymal stem cells or CA-MSCs (*ALDH1A3*), myofibroblasts (*ACTA2*), and fibroblasts (*COL1A1*). (iv) Endothelial cells (*CDH5*). (v) Contaminating cells (other than any of the listed above), which were not included in the subsequent analysis. The final dataset contained 33,718 genes across 106,521 cells.

**High- and low-stromal tumor identification**. After cell-type assignment, we computed the stromal cell fraction by sample:

$$\text{stromal cell fraction} = \frac{\#(\text{CA} - \text{MSCs}) + \#(\text{fibroblasts}) + \#(\text{myofibroblasts})}{\#(\text{all cells})}.$$

Samples with a ≥0.15 stromal cell fraction formed the high-stromal group (6 samples), and samples with a <0.15 stromal cell fraction formed the low-stromal group (14 samples).

**Differential expression analysis of scRNA-seq data**. In each cell-type population, differentially expressed genes (DEGs) between high- and low-stromal groups were identified using the *FindMarkers* function from Seurat. The expression of each gene in the high- and low-stromal groups formed two expression vectors. A two-sided Wilcoxon signed-rank test was applied to compare the means of the two expression vectors for each gene. The resulting *p*-values were adjusted based on Bonferroni correction using all genes in the dataset. Adjusted *p*-values cutoffs for DEGs are listed in Table S2. Differential expression of cytokines and SPs were of special interest. A full list of cytokines and SPs considered in the study is listed in Table S3.

**Single-cell regulatory network inference and clustering (pySCENIC)**. SCENIC[27] is a computational framework that predicts TF activities from scRNA-seq data. For each cell type, we inferred cell-specific TF activities using the Python implementation of SCENIC, pySCENIC (version 0.10.4)[26], with default parameters. We used the cis-regulatory DNA-motif database (hg38__refseq-r80__10kb_up_and_down_tss.mc9nr.feather, downloaded from https://resources.aertslab.org/cistarget/). The pySCENIC cellular regulon enrichment matrix (a.k.a. the AUC matrix) is the inferred TF activities by cell. Cutoffs for displaying gene-TF pairs are listed in Table S4.

**Pathway analysis (PROGENy)**. Pathway activities were inferred using the R package PROGNEy (version 1.12.0)[67] with default parameters. After inference, for each pathway signaled in each cell type, a two-sided Wilcoxon signed-rank test was applied between the high- and low-stromal groups. The resulting *p*-values are adjusted based on Bonferroni correction using all pathways in the dataset.

**Assessment of ligand–receptor interactions (CellPhoneDB)**. We used CellPhoneDB (version 3.1.0)[68] to identify the potential ligand–receptor interactions for each cell type based separately on the raw count matrices of the high- and low-stromal groups. For each group, the means of the average expression levels of interacting ligands in the sender population and interacting receptors in the receiver population were computed, and a one-sided Wilcoxon signed-rank test was used to assess the statistical significance of each interaction score. Cutoffs for ligand–receptor pairs are listed in Table S5.

**Spatial transcriptomics (ST) data analysis**. The ST dataset comprised 12 samples from one dataset[3], and 11 high-quality samples were selected for the subsequent analysis. Each raw count matrix was loaded as a Seurat object to include features detected in at least three cells and cells with at least 400 features. For each Seurat object, we applied SCTransform normalization followed by PCA. A total of 30 PCs were used in UMAP dimensional reduction. Based on the average expression of stromal cell markers (*COL1A1* and *ACTA2*), the samples were divided into the high-stromal group (*n* = 5) and the low-stromal group (*n* = 6). For each sample, we used the Seurat function *FindSpatiallyVariableFeatures* to identify features whose variability in expression was explained to some degree by spatial location, and we used the Moran's *I* test[69] to compute the spatial autocorrelation of each gene.

**Survival analysis**. We used R package MIND (Version 0.3.2)[22] to estimate the stromal cell fraction for each TCGA-OvCa sample. Specifically, the count matrix of the integrated scRNA-seq data was used as the prior cell-type specific profile. To improve the efficiency and accuracy of the estimates, we reassigned cells in the prior profiles to one of the four cell types: immune cells, stromal cells, endothelial cells, or ECCs. However, only common genes of the bulk and scRNA-seq datasets were considered in the deconvolution (16,428 genes). We used the CPM (counts per million) and log1p normalized TCGA-OvCa bulk raw count matrix as the expression to be deconvoluted. The outputs of the MIND *bMIND2* function included an estimated sample-specific cell-type fraction matrix.

Intersecting samples (*n* = 293) of TCGA-OvCa RNA-seq and clinical data were used in the survival analysis. Samples with an estimated >0.25 stromal cell fraction or with *COL1A1* raw count >20,000 formed the high-stromal group (28 samples), and samples with an estimated <0.15 stromal cell fraction formed the low-stromal group (220 samples). A Cox proportional hazards regression model was fit against the groups, and Kaplan–Meier survival curves were drawn with the R package survminer (Version 0.4.9) (https://CRAN.R-project.org/package=survminer).

**Peripheral blood monocyte isolation, CD8$^+$ T cell isolation, and macrophage differentiation**. PBMCs were freshly isolated by density gradient centrifugation using Ficoll Paque Plus (Sigma-

Aldrich) for 50 min at $400 \times g$[70]. Human Buffy Coat samples purchased from Vitalant fulfilled the exempt criteria of 45 CFR 46.101(b) (4) in accordance with the University of Pittsburgh guidelines. Monocytes were then isolated with CD14+ microbeads (MACS Miltenyi) and incubated for 5 days in RPMI/10% fetal calf serum medium and 1% penicillin/streptomycin solution (Sigma) supplemented with 25 ng/ml human macrophages colony-stimulating factor (M-CSF) (R&D Systems) for 4 days to stimulate macrophage differentiation. Macrophages were washed with PBS and cultured, as described below.

**Cell cultures.** The human ovarian cancer cell line OvCAR3 was purchased from ATCC and cultured in Roswell Park Memorial Institute (RPMI) 1640 medium supplemented with 10% heat-inactivated fetal bovine serum, penicillin (100 U/ml), and streptomycin (100 mg/ml). Normal MSCs or cancer-associated MSCs (CA-MSCs) were kindly donated by Dr. Coffman's lab. Briefly, normal MSCs were derived from surgical fallopian tubes of women undergoing surgery for benign indications, whereas CA-MSC were isolated from human HGSOC ascites as indicated previously[71]. Cells were cultured in Human Mesenchymal Stem Cell Growth Medium (ATCC). Cells were regularly tested for *Mycoplasma* contamination.

In the coculture experiments, OvCAR3 cells and MSC were plated alone or together at a 1:1 ratio ($0.5 \times 10^6$ cells). Co-cultures were maintained in RPMI and stem cell medium (1:1 ratio). After 7 days, the CM was collected and saved at $-80\,°C$.

**Cytokine arrays.** Macrophages were cultured with CM from OvCAR3 tumor cells (TC), MSC, or TC/MSC co-culture. After 48 h, macrophages were washed with PBS and cultured for 24 h. Supernatants were collected for a cytokine array assay (Raybiotech, AAH-CYT-5) performed according to the manufacturer's instructions. Membranes were developed, and the dots were quantified using Image J.

**CD8+ T cells activation and cytotoxic assay.** CD8+ T cells were isolated from fresh PBMC using CD8+ microbeads (MACS Miltenyi) following the manufacturer's instructions. Cells were activated with 30 U/mL rIL-2 and Dynabeads Human T-activator CD3/CD28 (ThermoFisher) and cultured with the CM of CA-MSCs in the presence or not of CXCR4 Ab (R&D Systems). CD8+ T cells stimulated only with IL-2 and CD3/CD28 microbeads were used as positive control. In indicated experiments, cells were stimulated with 200 ng/mL recombinant CXCL12. Following stimulation, CD8+ T cell supernatants were collected and analyzed for cytokine production by ELISA as described below.

**Enzyme-linked immunosorbent assay.** IFN-γ and granzyme B concentrations in the supernatants of CD8+ T cells were measured using a mouse ELISA kit (R&D Systems) following the manufacturer's protocol. IFN-γ and granzyme B concentrations were within the range of the standard curve. All points were done in triplicate, and the experiments were repeated three times. Samples were read in a microplate reader (Infinite 200 PRO, Tecan).

**Immunohistochemistry.** Slides were deparaffinized by baking overnight at 59 °C, and antigens were retrieved by heating in 0.1% citrate buffer for 10 min at 850 V in a microwave oven. For double-staining immunohistochemistry (IHC), we used Imm-PRESS duet staining HRP/AP polymer kits with anti-rabbit IgG-brown and anti-mouse IgG red (MP-7714, Vector Laboratories) according to the manufacturer's protocol. Nonspecific binding

sites were blocked with horse serum. The reactions with anti-CXCL12 (clone, Thermo Fisher) and anti-CD8 were for 16 h at 4 °C. Histology sections were observed using a Leica DM4 microscope. Images were acquired using a Leica DFC7000T camera and Leica Application Suite X.

**Statistics and reproducibility.** Statistical analysis and visualization were performed in R. The statistical methods used for each analysis are described within the texts and figure legends. Statistical significance is coded by the following symbols. *p*-value < 0.1, \**p*-value < 0.05, \*\**p*-value < 0.01, and \*\*\**p*-value < 0.001.

Graphs were generated using R packages: ggplot2 (version 3.3.6), ggpubr (version 0.4.0), ggrepel, circlize (version 0.9.1)[72], ComplexHeatmap (version 2.6.2)[73], and inlmisc (version 0.5.5). All violin plots report the 25% (lower hinge), 50%, and 75% quantiles (upper hinge) and the kernel density estimates as computed by the *geom_density* function of ggplot2 as the width. All boxplots report 25% (lower hinge), 50%, and 75% quantiles (upper hinge). The lower (upper) whiskers indicate the smallest (largest) observation greater (less) than or equal to the lower (upper) hinge $-1.5\times$ interquartile range (IQR) $(+1.5\times$ IQR) as default in the *geom_boxplot* function ggplot2.

## Data availability

In-house single-cell RNA sequencing data are deposited in the Gene Expression Omnibus (Accession No. GSE232314). Expression data from previous studies were obtained from the Gene Expression Omnibus (GEO) dataset and the European Genome-phenome Archive (EGA) dataset, under accession numbers GSE173682[19] (Regner et al.), GSE184880[20] (by Xu et al.), GSE189843[3] (by Stur et al.), and EGAS00001004935[14] (by Hornburg et al.). The selected studies and samples are listed in Table S1. The source data behind the graphs in the paper are available in Supplementary Data 1.

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

## Acknowledgements

The authors thank the authors from published studies for sharing their data on ovarian cancer scRNA-seq and spatial transcriptomics profiling as well as Cynthia Adams for preparing in-house scRNA-seq libraries. Data analysis was supported by the University of Pittsburgh Center for Research Computing and the Extreme Science and Engineering Discovery Environment (XSEDE), which is supported by National Science Foundation grant number OCI-1053575. Specifically, we used the Bridges2 system, which is supported by NSF award number ACI-1445606 at the Pittsburgh Supercomputing Center. This work was funded by the National Institutes of Health (R35GM146989 and R00CA207871) to H.U.O., Early Investigator Award-Ovarian Cancer Research Alliance Foundation and National Cancer Institute Grant R01CA276279 to S.C., Ovarian Cancer Research Alliance to R.B. and the National Natural Science Foundation of China (12101342) to L.Z. Funding for the open access charge was from the National Institutes of Health. This research was also supported in part by the University of Pittsburgh Center for Research Computing, RRID:SCR_022735, through the resources they provided. Specifically, this work used the H2P cluster, which is supported by NSF award number OAC-2117681.

## Author contributions

Conceptualization: H.U.O., S.C. and R.B. Methodology: H.U.O., L.Z. and S.C. Investigation: H.U.O., L.Z. and S.C. Visualization: L.Z. Supervision: H.U.O. Writing—original draft: H.U.O., L.Z. and S.C. Writing—review & editing: S.C. and R.B.

## Competing interests

The authors declare no competing interests.
