## [Peer Review File · Communications Biology]

Reviewers' comments:

Reviewer #1 (Remarks to the Author):

In the present study, Zhang et al. use scRNA-seq to address the dynamic features of transcriptomic data and integrate and validate the results from public dataset. Overall, the article is well written and data are nicely present. The dataset they provide will be beneficial for the field to study the TME of ovarian cancer. However, authors should put more efforts to describe the correlations between their observations and published works. Although I am positive for this manuscript, the following will be some of my suggestions that could be potentially improved this manuscript.

- Line200-201: Should be Fig.3 E and 3F. CXCL13 not CXCL3. Please also speculate why these chemokines are induced after MSC added into tumor cell lines.
- Line 263-267: The CXCL12hi locations are not coincident as COL1A2 and ACTA2. Please also explain why CXCL12 can restrain CD8 T cells and NK cells trafficking. The spatial data from public dataset are not really support the authors' analysis.

Reviewer #2 (Remarks to the Author):

This paper aims to profile high-grade serous ovarian cancer by profiling the tumour microenvironment. The study compares two experimental samples and attempts to validate the findings in publicly available datasets to determine the robustness of the findings.

Specific comments:

1. The abstract is non-descript with regard to the experimental arm of the study - this should be explained upfront
2. abstract requires limitations of the study
3. high stromal and low stromal are not defined - please define. How were these identified (pathology review of tissue?).
4. how many cells of each tumour were scRNA seq performed on?
5. high/low stromal defined in line 127 - this should be brought forward in the text.
6. line 182 - not clear why the authors would reference their own work in murine studies without an explanation? This is then repeated in the discussion. Please avoid repetition here.
7. I am missing the limitations of the study (small experimental groups), pairing of findings to public datasets.
8. This needs to be well explained in both the abstract and the discussion.

Response to the Editor

Comments	Responses
1. To make more efforts to describe the correlation between your own dataset of small sample size with the published work and the limitations of this study.	We have added more connections between our observations and the published works as outlined in our response. In addition we have added a review of some of the limitations of our datasets/study to the discussion. “Nonetheless, this study has limitations. First, the relatively small sample size imposes constraints on the flexibility of multivariable modeling. Given that our in-house cohort comprises only two tumor types, we attempted to address this limitation by integrating additional scRNA-seq datasets to capture the variability present across HGSOc tumors. To enable a joint analysis of both in-house and public datasets, we had to choose a data integration approach (for detailed information, please refer to the Methods section). It is important to note that each cohort entails a batch effect due to slight variations in sample processing and differences in scRNA-seq library preparation and sequencing. However, the integration of several datasets has the advantage of mitigating biases related to cell preparation and dissociation, such as the preferential liberation of specific cell types during tissue dissociation. Additionally, our ligand-receptor analysis relies on a limited database of known ligand-receptor interactions and can only infer potential communication between different cell types. Therefore, the ligand-receptor analysis should be considered primarily for hypothesis generation, and a more sensible future approach would involve conducting functional studies to delve into the molecular crosstalk. Further, our study lacks matching clinical data regarding treatment allocation and responses to treatment. This limitation is due to the retrospective nature of our analysis.”
2. To be more extensive in the spatial transcriptomic validation and to ensure the spatial transcriptomic data supports the analysis and the claims.	We have added additional analysis with spatial transcriptomic dataset and annotations to the figure to support our claims as outlined in our response below. Briefly, we computed the correlation of CXCL12 expression and COL1A1/ACTA2 expression by sample and found strong correlation for both pairs. This plot has been included as Figure 6E.

Point-by-point response to reviewers

Response to Reviewer #1

In the present study, Zhang et al. use scRNA-seq to address the dynamic features of transcriptomic data and integrate and validate the results from public dataset. Overall, the article is well written and data are nicely present. The dataset they provide will be beneficial for the field to study the TME of ovarian cancer.

Thank you very much for taking the time to review the manuscript and for providing feedback. We are confident that your comments have helped us to improve the manuscript. Please find below our point-to-point responses to your suggestions.

Comments	Responses
1. However, authors should put more efforts to describe the correlations between their observations and published works. Although I am positive for this manuscript, the following will be some of my suggestions that could be potentially improved this manuscript.	Thank you very much for raising this point. As the reviewer suggested, we have added more connections/citations between our observations and the published works as outlined below: Lines 333-340: We found that both CD8+ T cells and NK cells are abundant in CXCL12+ tumor area. This is consistent with previous studies in pancreatic cancer showing that CAF-secreted CXCL12 attract peripheral CXCR4+ CD8+ T cell towards activated CAFs located in the stromal regions surrounding the tumour^{41,43}. This leads to the sequestration of CD8+ T cells in the stromal compartment and reduced migration into tumour islets^{41,43}. In addition, preclinical studies showed that both pharmacological inhibition of CXCR4 and genetic ablation of CXCL12 producing CAFs led to a rapid accumulation of CD8+ T cells within the tumour and reduced tumour growth⁴¹.
2. Line 200-201: Should be Fig.3 E and 3F. CXCL13 not CXCL3. Please also speculate why these chemokines are induced after MSC	We apologize for the typo. We have fixed the typo. We have also added a paragraph (lines 317-323) describing the potential mechanisms of MSC-secreted factors control chemokine expression in macrophages as described above. “These cytokines are known to induce proliferation, migration and metastasis of cancer cells (31, 32). MSCs induce the recruitment of macrophages into the tumors and convert them into a M2-like signature via secretion of inflammatory factors, including IDO, IL-10, IL-6 PEG2, TGF-beta, and TNF-alpha (33, 34). The expression of CXCL13 is mainly induced by TNF-a and IL-10 (35, 36) whereas CXCL1 and CXCL5 expression levels

added into tumor cell lines.	are modulated by TNF-α (32, 37). Therefore, a potential mechanism of CAMSCs to induce the expression CXCL1, CXCL5 and CXCL13 in macrophages is via secretion of cytokines and chemokines, including TNF-α, IL-10 and TGF-β”.
3. Line 263-267: The CXCL12hi locations are not coincident as COL1A2 and ACTA2. Please also explain why CXCL12 can restrain CD8 T cells and NK cells trafficking. The spatial data from public dataset are not really support the authors’ analysis.	We computed the correlation of CXCL12 expression and COL1A1/ACTA2 expression by sample and found strong correlation for both pairs as seen in Figure R1. This plot has been included as Figure 6E. “Additionally, we computed the Spearman correlation between expression of CXCL12 and COL1A1 and ACTA2 across samples and we observed higher correlation between CXCL12 and COL1A1 and ACTA2 in high-stromal samples (Fig. 6E).”  Figure R1: Heatmap showing the Spearman correlation coefficients between the expression of COL1A1/ACTA2 and CXCL12.

Figure R2: (A) Representative spatial feature plots showing mRNA levels of stromal markers (COL1A1, ACTA2, EPCAM), CXCL12, CD8+ T cell marker (CD8A) and NK marker (KLRB1) from two high-stromal samples (B) and two low-stromal samples. Arrows indicate regions COL1A1, ACTA2 and CXCL12.

To help the readers visualize the correlation of CXCL12 and COL1A1/ACTA2, we have included two more samples in **Figure R2**, and have added the arrows where those genes are expressed.

We have also added in the main text the explanation why CXCL12 can restrain CD8 T cells and NK cells trafficking:

“We found that both CD8+ T cells and NK cells are abundant in CXCL12+ tumor area. This is consistent with previous studies in pancreatic cancer showing that CAF-secreted CXCL12 attract peripheral CXCR4+ CD8+ T cell towards activated CAFs located in the stromal regions surrounding the tumour. This leads to the sequestration of CD8+ T cells in the stromal compartment and reduced migration into tumour islets (41, 43). In addition, preclinical studies showed that both pharmacological inhibition of CXCR4 and genetic ablation of CXCL12 producing CAFs led to a rapid accumulation of CD8+ T cells within the tumour and reduced tumour growth (41)”.

Response to Reviewer #2

This paper aims to profile high-grade serous ovarian cancer by profiling the tumour microenvironment. The study compares two experimental samples and attempts to validate the findings in publicly available datasets to determine the robustness of the findings.

Thank you very much for taking the time to review the manuscript and for providing feedback. We are confident that your comments have helped us to improve the manuscript. Please find below our point-to-point responses to your suggestions.

Comments	Responses
1. The abstract is non-descript with regard to the experimental arm of the study - this should be explained upfront. . . . Abstract requires limitations of the study.	Thank you very much for raising this point. We have edited our abstract, withing the text limitations, to address both of these comments as indicated below. “We integrated single-cell transcriptomics of the HGSOC TME from public and in-house datasets (n=20) and stratified tumors based upon high vs. low stromal cell content. Although our cohort size was small, our data suggests a distinct transcriptomic landscape for immune and non-immune cells in high-stromal vs. low-stromal tumors. . . . Analysis of cell-cell communication indicated that epithelial cancer cells and CA-MSCs secreted CXCL12 that interacted with the CXCR4 receptor, which was overexpressed on NK and CD8⁺ T cells. Dual IHC staining showed that tumor infiltrating CD8 T cells localized in proximity of CXCL12+ tumor area. Moreover, CXCL12 and/or CXCR4 antibodies confirmed the immunosuppressive role of CXCL12-CXCR4 in high-stromal tumors.”
2. High stromal and low stromal are not defined - please define. How were these identified (pathology review of tissue?).	We apologize for not making this clear. We defined high stromal and low stromal according to stromal cell type abundance. We did not have access to the pathological review of samples. “We performed scRNA-seq to characterize the malignant, immune, and stromal cells associated with two treatment-naïve HGSOC samples—one high-stromal and one low-stromal according to stromal cell type abundance (see MATERIALS AND METHODS). “High- and low-stromal tumor identification. After cell type assignment, we computed the stromal cell fraction by sample:

	stromal cell fraction $= \frac{\#(\text{CA-MSCs}) + \#(\text{fibroblasts}) + \#(\text{myofibroblasts})}{\#(\text{all cells})}$ Samples with a ≥ 0.15 stromal cell fraction formed the high-stromal group (6 samples), and samples with a < 0.15 stromal cell fraction formed the low-stromal group (14 samples)."																																																
3. How many cells of each tumour were scRNA seq performed on?	We have included an additional column in Table S1 to list the number of cells used in the study by sample in Table R1. Table R1: Number of cells/spots in each sample.    Sample ID #cells Sample ID #cells used in this study Sample ID #spots     Pitt_1 7,612 Hornburg_1 7,022 Stur_01 428   Pitt_2 10,780 Hornburg_2 14,589 Stur_02 285   Regner_1 6,147 Hornburg_3 1,864 Stur_03 172   Regner_2 5,966 Hornburg_4 3,660 Stur_04 170   Xu_1 6,421 Hornburg_5 3,711 Stur_05 596   Xu_2 3,026 Hornburg_6 4,823 Stur_06 347   Xu_3 3,537 Hornburg_7 1,429 Stur_07 499   	Sample ID	#cells	Sample ID	#cells used in this study	Sample ID	#spots	Pitt_1	7,612	Hornburg_1	7,022	Stur_01	428	Pitt_2	10,780	Hornburg_2	14,589	Stur_02	285	Regner_1	6,147	Hornburg_3	1,864	Stur_03	172	Regner_2	5,966	Hornburg_4	3,660	Stur_04	170	Xu_1	6,421	Hornburg_5	3,711	Stur_05	596	Xu_2	3,026	Hornburg_6	4,823	Stur_06	347	Xu_3	3,537	Hornburg_7	1,429	Stur_07	499
Sample ID	#cells	Sample ID	#cells used in this study	Sample ID	#spots																																												
Pitt_1	7,612	Hornburg_1	7,022	Stur_01	428																																												
Pitt_2	10,780	Hornburg_2	14,589	Stur_02	285																																												
Regner_1	6,147	Hornburg_3	1,864	Stur_03	172																																												
Regner_2	5,966	Hornburg_4	3,660	Stur_04	170																																												
Xu_1	6,421	Hornburg_5	3,711	Stur_05	596																																												
Xu_2	3,026	Hornburg_6	4,823	Stur_06	347																																												
Xu_3	3,537	Hornburg_7	1,429	Stur_07	499																																												
4. High/low stromal defined in line 127 - this should be brought forward in the text.	As we noted above, we revised our first sentence as below: "We performed scRNA-seq to characterize the malignant, immune, and stromal cells associated with two treatment-naïve HGSOc samples—one high-stromal and one low-stromal according to stromal cell type abundance (see MATERIALS AND METHODS and below)."																																																
5. Line 182 - not clear why the authors would reference their own work in murine studies without an explanation? This is then repeated in the discussion. Please avoid repetition here.	To avoid confusion, we have removed the sentence and the reference of our previous work. However, we have kept this reference in the discussion. In fact, our previous study showed that stromal cells regulate tumor infiltrating myeloid cell phenotype and function in a murine model of OvCa supporting the findings and analyses of this manuscript.																																																
6/7/. I am missing the limitations of the study (small experimental groups), pairing of findings to public datasets. This needs to be well	Thank you very much for raising this point. We have added limitations of our study in the discussion section. "Nonetheless, this study has limitations. First, the relatively small sample size imposes constraints on the flexibility of multivariable modeling. Given that our in-house cohort comprises only two tumor types, we attempted to address this limitation by integrating																																																

explained in both the abstract and the discussion.	additional scRNA-seq datasets to capture the variability present across HGSOc tumors. To enable a joint analysis of both in-house and public datasets, we had to choose a data integration approach (for detailed information, please refer to the Methods section). It is important to note that each cohort entails a batch effect due to slight variations in sample processing and differences in scRNA-seq library preparation and sequencing. However, the integration of several datasets has the advantage of mitigating biases related to cell preparation and dissociation, such as the preferential liberation of specific cell types during tissue dissociation. Additionally, our ligand-receptor analysis relies on a limited database of known ligand-receptor interactions and can only infer potential communication between different cell types. Therefore, the ligand-receptor analysis should be considered primarily for hypothesis generation, and a more sensible future approach would involve conducting functional studies to delve into the molecular crosstalk. Further, our study lacks matching clinical data regarding treatment allocation and responses to treatment. This limitation is due to the retrospective nature of our analysis."
---	---

References

- Amiri, K. I. and A. Richmond (2003). "Fine tuning the transcriptional regulation of the CXCL1 chemokine." Prog Nucleic Acid Res Mol Biol **74**: 1-36.
- Cascio, S., et al. (2021). "Cancer-associated MSC drive tumor immune exclusion and resistance to immunotherapy, which can be overcome by Hedgehog inhibition." Sci Adv **7**(46): eabi5790.
- Feig, C., et al. (2013). "Targeting CXCL12 from FAP-expressing carcinoma-associated fibroblasts synergizes with anti-PD-L1 immunotherapy in pancreatic cancer." Proc Natl Acad Sci U S A **110**(50): 20212-20217.
- Freeman, P. and A. Mielgo (2020). "Cancer-Associated Fibroblast Mediated Inhibition of CD8+ Cytotoxic T Cell Accumulation in Tumours: Mechanisms and Therapeutic Opportunities." Cancers (Basel) **12**(9).

REVIEWERS' COMMENTS:

Reviewer #1 (Remarks to the Author):

I am satisfied with the edit and have no further questions on the manuscript.

Reviewer #2 (Remarks to the Author):

the authors have now addressed my concerns. Sample size remains a challenge but these initial spatial studies are often descriptive because of the costs associated with these assays. The authors have acknowledged this appropriately.